# Automatically Generating Rhythmic Verse with Neural Networks

## Abstract

We propose two novel methodologies for the automatic generation of rhythmic poetry in a variety of forms. The first approach uses a neural language model trained on a phonetic encoding to learn an implicit representation of both the *form* and *content* of English poetry. This model can effectively learn common poetic devices such as rhyme, rhythm and alliteration. The second approach considers poetry generation as a constraint satisfaction problem where a generative neural language model is tasked with learning a representation of content, and a discriminative weighted finite state machine constrains it on the basis of form. By manipulating the constraints of the latter model, we can generate coherent poetry with arbitrary forms and themes. A large-scale extrinsic evaluation demonstrated that participants consider machine-generated poems to be written by humans 54% of the time. In addition, participants rated a machine-generated poem to be the best amongst all evaluated.

## 1 Introduction

Poetry is an advanced form of linguistic communication, in which a message is conveyed that satisfies both aesthetic and semantic constraints. As poetry is one of the most expressive forms of language, the *automatic* creation of texts recognisable as poetry is difficult. In addition to requiring an understanding of many aspects of language including phonetic patterns such as rhyme, rhythm and alliteration, poetry composition also requires a deep understanding of the meaning of language.

Poetry generation can be divided into two sub-tasks, namely the problem of *content*, which is concerned with a poem's semantics, and the problem of *form*, which is concerned with the aesthetic rules that a poem follows. These rules may describe aspects of the literary devices used, and are usually highly prescriptive. Examples of different forms of poetry are limericks, ballads and sonnets. Limericks, for example, are characterised by their strict rhyme scheme (AABBA), their rhythm (two unstressed syllables followed by one stressed syllable) and their shorter third and fourth lines. Creating such poetry requires not only an understanding of the language itself, but also of how it sounds when spoken aloud.

Statistical text generation usually requires the construction of a generative language model that explicitly learns the probability of any given word given previous context. Neural language models (Schwenk and Gauvain, 2005; Bengio et al., 2006) have garnered signficant research interest for their ability to learn complex syntactic and semantic representations of natural language (Mikolov et al., 2010; Sutskever et al., 2014; Cho et al., 2014; Kim et al., 2015). Poetry generation is an interesting application, since performing this task automatically requires the creation of models that not only focus on *what* is being written (content), but also on *how* it is being written (form).

We experiment with two novel methodologies for solving this task. The first involves training a model to learn an implicit representation of content and form through the use of a phonological encoding. The second involves training a generative language model to represent content, which is then constrained by a discriminative pronunciation model, representing form. This second model is of particular interest because poetry with arbitrary rhyme, rhythm, repetition and themes can be generated by tuning the pronunciation model.

## 2 Related Work

Automatic poetry generation is an important task due to the significant challenges involved. Most systems that have been proposed can loosely be categorised as rule-based expert systems, or statistical approaches.

Rule-based poetry generation attempts include case-based reasoning (Gervás, 2000), template-based generation (Colton et al., 2012), constraint satisfaction (Toivanen et al., 2013) and text mining (Netzer et al., 2009). These approaches are often inspired by how humans might generate poetry.

Statistical approaches, conversely, make no assumptions about the creative process. Instead, they attempt to extract statistical patterns from existing poetry corpora in order to construct a language model, which can then be used generate new poetic variants (Yi et al., 2016; Greene et al., 2010). The work of Zhang and Lapata (2014) is similar to ours, in that they make use of neural language models. For the task of automatic generation of classical Chinese poetry, they were able to outperform all other Chinese poetry generation systems with both manual and automatic evaluation.

## 3 Phonetic-level Model

Our first model is a pure neural language model, trained on a phonetic encoding of poetry in order to represent both form and content. Phonetic encodings of language represent information as sequences of around 40 basic acoustic symbols. Training on phonetic symbols allows the model to learn effective representations of pronunciation, including rhyme and rhythm.

However, just training on a large corpus of poetry data is not enough. Specifically, two problems need to be overcome. 1) Phonetic encoding results in information loss: words that have the same pronunciation (homophones) cannot be perfectly reconstructed from the corresponding phonemes. This means that we require an additional probabilistic model in order to determine the most likely word given a sequence of phonemes. 2) The variety of poetry and poetic devices one can use—e.g., rhyme, rhythm, repetition—means that poems sampled from a model trained on *all* poetry would be unlikely to maintain internal consistency of meter and rhyme. It is therefore important to train the model on poetry which has its own internal consistency.

Thus, the model comprises three steps: transliterating an ortographic sequence to its phonetic representation, training a neural language model on the phonetic encoding, and decoding the generated sequence back from phonemes to orthographic symbols.

**Phonetic encoding** To solve the first step, we apply a combination of word lookups from the CMU pronunciation dictionary (Weide, 2005) with letter-to-sound rules for handling out-of-vocabulary words. These rules are based on the CART techniques described by Black et al. (1998), and are represented with a simple Finite State Transducer[1]. The number of letters and number of phones in a word are rarely a one-to-one match: letters may match with up to three phones. In addition, virtually all letters can, in some contexts, map to zero phones, which is known as 'wild' or epsilon. Expectation Maximisation is used to compute the probability of a single letter matching a single phone, which is maximised through the application of Dynamic Time Warping (Myers et al., 1980) to determine the most likely position of epsilon characters.

Although this approach offers full coverage over the training corpus—even for abbreviated words like *ask'd* and archaic words like *renewest*—it has several limitations. Irregularities in the English language result in difficulty determining general letter-to-sound rules that can manage words with unusual pronunciations such as "colonel" and "receipt" [2].

In addition to transliterating words into phoneme sequences, we also represent word break characters as a specific symbol. This makes decipherment, when converting back into an orthographic representation, much easier. Phonetic transliteration allows us to construct a phonetic poetry corpus comprising 1,046,536 phonemes.

**Neural language model** We train a Long-Short Term Memory Network (Hochreiter and Schmidhuber, 1997) on the phonetic representation of our poetry corpus. The model is trained using stochastic gradient descent to predict the next phoneme given a sequence of phonemes. Specifically, we

---

[1]Implemented using FreeTTS (Walker et al., 2010)

[2]An evaluation of models in American English, British English, German and French was undertaken by Black et al. (1998), who reported an externally validated per token accuracy on British English as low as 67%. Although no experiments were carried out on corpora of early-modern English, it is likely that this accuracy would be significantly lower.

maximize a multinomial logistic regression objective over the final softmax prediction. Each phoneme is represented as a 256-dimensional embedding, and the model consists of two hidden layers of size 256. We apply backpropagation-through-time (Werbos, 1990) for 150 timesteps, which roughly equates to four lines of poetry in sonnet form. This allows the network to learn features like rhyme even when spread over multiple lines. Training is preemptively stopped at 25 epochs to prevent overfitting.

**Orthographic decoding**   When decoding from phonemes back to orthographic symbols, the goal is to compute the most likely word corresponding to a sequence of phonemes. That is, we compute the most probable hypothesis word $W$ given a phoneme sequence $\rho$:

$$arg\,max_i\,P\,(\,W_i\,|\,\rho\,) \qquad (1)$$

We can consider the phonetic encoding of plaintext to be a *homophonic cipher*; that is, a cipher in which each symbol can correspond to one or more possible decodings. The problem of homophonic decipherment has received significant research attention in the past; with approaches utilising Expectation Maximisation (Knight et al., 2006), Integer Programming (Ravi and Knight, 2009) and A* search (Corlett and Penn, 2010).

Transliteration from phonetic to an orthographic representation is done by constructing a Hidden Markov Model using the CMU pronunciation dictionary (Weide, 2005) and an n-gram language model. We calculate the transition probabilities (using the n-gram model) and the emission matrix (using the CMU pronunciation dictionary) to determine pronunciations that correspond to a single word. All pronunciations are naively considered equiprobable. We perform Viterbi decoding to find the most likely sequence of words. This means finding the most likely word $w_{t+1}$ given a previous word sequence $(w_{t-n}, ..., w_t)$.

$$arg\,max_{w_{t+1}}\,P\,(\,w_{t+1}\,|\,w_1,\,...,\,w_t\,) \qquad (2)$$

If a phonetic sequence does not map to any word, we apply the heuristic of artificially breaking the sequence up into two subsequences at index $n$, such that $n$ maximises the n-gram frequency of the subsequences.

And humble and their fit *flees* are wits size
but that one made and made thy step me lies

————————————————

Cool light the golden dark in any way
the birds a *shade* a laughter turn away

————————————————

Then adding wastes retreating white as thine
She watched what eyes are breathing awe what shine

————————————————

But sometimes shines so covered how the beak
Alone in pleasant skies no more to seek

Figure 1: Example output of the phonetic-level model trained on Iambic Pentameter poetry (grammatical errors are emphasised).

**Output**   A popular form of poetry with strict internal structure is the sonnet. Popularised in English by Shakespeare, the sonnet is characterised by a strict rhyme scheme and exactly fourteen lines of Iambic Pentameter (Greene et al., 2010). Since the 17,134 word tokens in Shakespeare's 153 sonnets are insufficient to train an effective model, we augment this corpus with poetry taken from the website *sonnets.org*, yielding a training set of 288,326 words and 1,563,457 characters.

An example of the output when training on this sonnets corpus is provided in Figure 1. Not only is it mostly in strict Iambic Pentameter, but the grammar of the output is mostly correct and the poetry contains rhyme.

# 4   Constrained Character-level Model

As the example shows, phonetic-level language models are effective at learning poetic form, despite small training sets and relatively few parameters. However, the fact that they require training data with internal poetic consistency implies that they do not generalise to other forms of poetry. That is, in order to generate poetry in Dactylic Hexameter (for example), a phonetic model must be trained on a corpus of Dactylic poetry. Not only is this impractical, but in many cases no corpus of adequate size even exists. Even when such poetic corpora are available, a new model must be trained for each type of poetry. This precludes tweaking the form of the output, which is important when generating poetry automatically.

We now explore an alternative approach. Instead of attempting to represent both form and content in a single model, we construct a pipeline containing a generative language model represent-

ing *content*, and a discriminative model representing *form*. This allows us to represent the problem of creating poetry as a constraint satisfaction problem, where we can modify constraints to restrict the types of poetry we generate.

**Character Language Model**   Rather than train a model on data representing features of *both* content and form, we now use a simple character-level model (Sutskever et al., 2011) focused solely on content. This approach offers several benefits over the word-level models that are prevalent in the literature. Namely, their more compact vocabulary allows for more efficient training; they can learn common prefixes and suffixes to allow us to sample words that are not present in the training corpus and can learn effective language representations from relatively small corpora; and they can handle archaic and incorrect spellings of words.

As we no longer need the model to explicitly represent the form of generated poetry, we can loosen our constraints when choosing a training corpus. Instead of relying on poetry only in sonnet form, we can instead construct a generic corpus of poetry taken from online sources. This corpus is composed of 7.56 million words and 34.34 million characters, taken largely from $20^{th}$ Century poetry books found online. The increase in corpus size facilitates a corresponding increase in the number of permissible model parameters. This allows us to train a 3-layer LSTM model with 2048-dimensional hidden layers. The model was trained to predict the next character given a sequence of characters, using stochastic gradient descent. We attenuate the learning rate over time, and by 20 epochs the model converges.

**Rhythm Modeling**   Although a character-level language model trained on a corpus of generic poetry allows us to generate interesting text, internal irregularities and noise in the training data prevent the model from learning important features such as rhythm. Hence, we require an additional classifier to constrain our model by either accepting or rejecting sampled lines based on the presence or absence of these features. As the presence of *meter* (rhythm) is the most characteristic feature of poetry, it therefore must be our primary focus.

Pronunciation dictionaries have often been used to determine the syllabic stresses of words (Colton et al., 2012; Manurung et al., 2000; Misztal and Indurkhya, 2014), but suffer from some limitations

for constructing a classifier. All word pronunciations are considered equiprobable, including archaic and uncommon pronunciations, and pronunciations are provided context free, despite the importance of context for pronunciation[3]. Furthermore, they are constructed from American English, meaning that British English may be misclassified.

These issues are circumvented by applying lightly supervised learning to determine the contextual stress pattern of any word. That is, we exploit the latent structure in our corpus of sonnet poetry, namely, the fact that sonnets are composed of lines in rigid Iambic Pentameter, and are therefore exactly ten syllables long with alternating syllabic stress. This allows us to derive a *syllable-stress distribution*. Although we use the sonnets corpus for this, it is important to note that any corpus with such a latent structure could be used.

By representing a line as a cascade of Weighted Finite State Transducers (WFST), we can perform Expectation Maximisation over the poetry corpus to obtain a probabilistic classifier which enables us to determine the most likely stress patterns for each word. Every word is represented by a single transducer. Since weights can be assigned to state transitions, we can model the probability that a given input string maps to a particular output.

In each cascade, a sequence of input words is mapped onto a sequence of stress patterns $\langle \times, / \rangle$ where each pattern is between 1 and 5 syllables in length[4]. We initially set all transition probabilities equally, as we make no assumptions about the stress distributions in our training set. We then iterate over each line of the sonnet corpus, using Expectation Maximisation to train the cascades. In practice, there are several de facto variations of Iambic meter which are permissible, as shown in Figure 2. We train the rhythm classifier by converging the cascades to whatever output is the most likely given the line.

**Constraining the model**   To generate poetry using this model, we sample sequences of characters from the character-level language model. To impose rhythm constrains on the language model, we first represent these sampled characters at the word level and pool sampled characters into word

---

[3]For example, the independent probability of stressing the single syllable word *at* is 40%, but this increases to 91% when the following word is *the* (Greene et al., 2010)

[4]Words of more than 5 syllables comprise less than 0.1% of the lexicon (Aoyama and Constable, 1998).

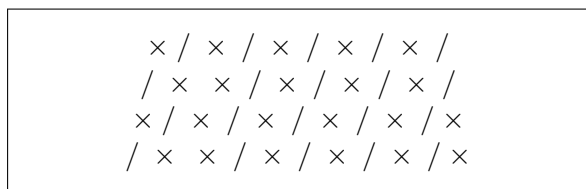

Figure 2: Permissible variations of Iambic Pentameter in Shakespeare's sonnets.

tokens in an intermediary buffer. We then apply the separately trained word-level WFSTs to construct a cascade of this buffer and perform Viterbi decoding over the cascade. This defines the distribution of stress-patterns over our word tokens.

We can represent this cascade as a probabilistic classifier, and accept or reject the buffered output based on how closely it conforms to the desired meter. While sampling sequences of words from this model, the entire generated sequence is passed to the classifier each time a new word is sampled. The pronunciation model then returns the probability that the entire line is within the specified meter. If a new word is rejected by the classifier, the state of the network is rolled back to the state of the last formulaically acceptable line, removing the rejected word from memory. The constraint on rhythm can be controlled by adjusting the acceptability threshold of the classifier. By increasing the threshold, output focuses on form over content. Conversely, decreasing the criterion puts greater emphasis on content.

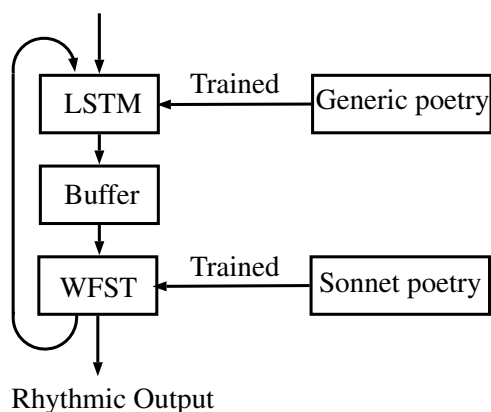

Rhythmic Output

### 4.1 Themes and Poetic devices

It is important for any generative poetry model to include themes and poetic devices. One way to achieve this would be by constructing a corpus that exhibits the desired themes and devices. To create a themed corpus about 'love', for instance,

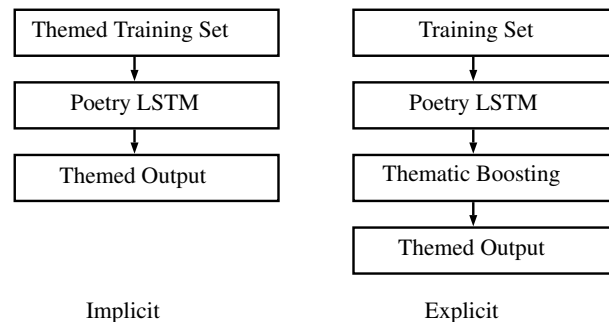

Implicit Explicit

Figure 3: Two approaches for generating themed poetry.

we would aggregate love poetry to train the model, which would thus learn an implicit representation of love. However, this forces us to generate poetry according to discrete themes and styles from pretrained models, requiring a new training corpus for each model. In other words, we would suffer from similar limitations as with the phonetic-level model, in that we require a dedicated corpus. Alternatively, we can manipulate the language model by boosting character probabilities at sample time to increase the probability of sampling thematic words like 'love'. This approach is more robust, and provides us with more control over the final output, including the capacity to vary the inclusion of poetic devices in the output.

**Themes** In order to introduce thematic content, we heuristically boost the probability of sampling words that are semantically related to a theme word from the language model. First, we compile a list of similar words to a *key* theme word by retrieving its semantic neighbours from a distributional semantic model (Mikolov et al., 2013). For example, the theme *winter* might include thematic words *frozen*, *cold*, *snow* and *frosty*. We represent these semantic neighbours at the character level, and heuristically boost their probability by multiplying the sampling probability of such character strings by a function of their cosine similarity to the key word. Thus, the likelihood of sampling a thematically related word is artificially increased, while still constraining the model rhythmically.

**Poetic devices** A similar method may be used for poetic devices such as assonance, consonance and alliteration. Since these devices can be orthographically described by the repetition of identical sequences of characters, we can apply the

| Errors per line | 1 | 2 | 3 | 4 | Total |
|---|---|---|---|---|---|
| Phonetic Model | 11 | 2 | 3 | 1 | 28 |
| Character Model + WFST | 6 | 5 | 1 | 1 | 23 |
| Character Model | 3 | 8 | 7 | 7 | 68 |

Table 1: Number of lines with $n$ errors from a set of 50 lines generated by each of the three models.

| | Word | Line | Coverage |
|---|---|---|---|
| Wikipedia | 64.84% | 83.35% | 97.53% |
| Sonnets | 85.95% | 80.32% | 99.36% |

Table 2: Error when transliterating text into phonemes and reconstructing back into text.

same heuristic to boost the probability of sampling character strings that have previously been sampled. That is, to sample a line with many instances of alliteration (multiple words with the same initial sound) we record the historical frequencies of characters sampled at the beginning of each previous word. After a word break character, we boost the probability that those characters will be sampled again in the softmax. We only keep track of frequencies for a fixed number of time steps. By increasing or decreasing the size of this window, we can manipulate the prevalence of alliteration. Variations of this approach are applied to invoke consonance (by boosting intra-word consonants) and assonance (by boosting intra-word vowels). An example of two sampled lines with high degrees of alliteration, assonance and consonance is given in Figure 4c.

## 5 Evaluation

In order to examine how effective our methodologies for generating poetry are, we evaluate the proposed models in two ways. First, we perform an intrinsic evaluation where we examine the quality of the models and the generated poetry. Second, we perform an extrinsic evaluation where we evaluate the generated output using human annotators, and compare it to human-generated poetry.

### 5.1 Intrinsic evaluation

To evaluate the ability of both models to generate formulaic poetry that adheres to rhythmic rules, we compared sets of fifty sampled lines from each model. The first set was sampled from the phonetic-level model trained on Iambic poetry.

The second set was sampled from the character-level model, constrained to Iambic form. For comparison, and to act as a baseline, we also sampled from the *unconstrained* character model.

We created gold-standard syllabic classifications by recording each line spoken-aloud, and marking each syllable as either stressed or unstressed. We then compared these observations to loose Iambic Pentameter (containing all four variants), to determine how many syllabic misclassifications existed on each line. This was done by speaking each line aloud, and noting where the speaker put stresses.

As Table 1 shows, the constrained character level model generated the most formulaic poetry. Results from this model show that 70% of lines had zero mistakes, with frequency obeying an inverse power-law relationship with the number of errors. We can see that the phonetic model performed similarly, but produced more subtle mistakes than the constrained character model: many of the errors were single mistakes in an otherwise correct line of poetry.

In order to investigate this further, we examined to what extent these errors are due to transliteration (i.e., the phonetic encoding and orthographic decoding steps). Table 2 shows the reconstruction accuracy per word and per line when transliterating either Wikipedia or Sonnets to phonemes using the CMU pronunciation dictionary and subsequently reconstructing English text using the n-gram model[5]. Word accuracy reflects the frequency of *perfect* reconstruction, whereas per line tri-gram similarity (Kondrak, 2005) reflects the *overall* reconstruction. Coverage captures the percentage of in-vocabulary items. The relatively low per-word accuracy achieved on the Wikipedia corpus is likely due to the high frequency of out-of-vocabulary words. The results show that a significant number of errors in the phonetic-level model are likely to be caused by transliteration mistakes.

### 5.2 Extrinsic evaluation

We conducted an indistinguishability study with a selection of automatically generated poetry and human poetry. As extrinsic evaluations are expensive and the phonetic model was unlikely to do well (as illustrated in Figure 4e: the model generates good Iambic form, but not very good English),

---

[5]Obviously, calculating this value for the character-level model makes no sense, since no transliteration occurs in that case.

(a)
The crow crooked on more beautiful and free,
He journeyed off into the quarter sea.
his radiant ribs girdled empty and very -
least beautiful as dignified to see.

(c)
Man with the broken blood blue glass and gold.
Cheap chatter chants to be a lover do.

(e)
The son still streams and strength and spirit.
The ridden souls of which the fills of.

(b)
Is that people like things
(are the way we to figure it
out) and I thought of you
reading and then is your
show or you know we will
finish along will you play.

(d)
How dreary to be somebody,
How public like a frog
To tell one's name the livelong day
To an admiring bog.

Figure 4: Examples of automatically generated and human generated poetry. (a) Character-level model - Strict rhythm regularisation - Iambic - No Theme. (b) Character-level model - Strict rhythm regularisation - Anapest. (c) Character-level model - Boosted alliteration/assonance. (d) Emily Dickinson - I'm nobody, who are you? (e) Phonetic-level model - Nonsensical Iambic lines.

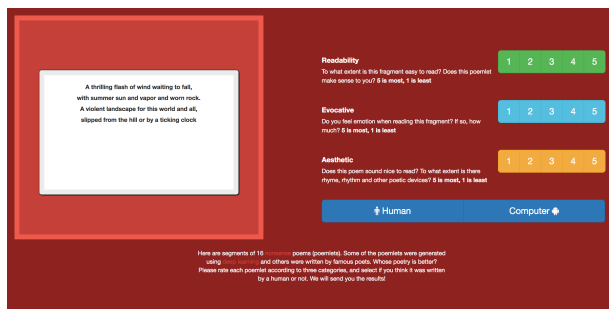

Figure 5: The experimental environment for asking participants to distinguish between automatically generated and human poetry.

we only evaluate on the constrained character-level model.

The aim of the study was to determine whether participants could distinguish between human and generated poetry, and if so to what extent. A set of 70 participants (of whom 61 were English native speakers) were each shown a selection of randomly chosen poetry segments, and were invited to classify them as either human or generated. Participants were recruited from friends and people within poetry communities, with an age range of 17 to 80, and a mean age of 29. Our participants were not financially incentivised, perceiving the evaluation as an intellectual challenge.

In addition to the classification task, each partic-

ipant was also invited to rate each poem on a 1-5 scale with respect to three criteria, namely readability, form and evocation (how much emotion did a poem elicit). We naively consider the overall quality of a poem to be the mean of these three measures. We used a custom web-based environment, built specifically for this evaluation[6], which is illustrated in Figure 5. Based on human judgments, we can determine whether the models presented in this work can produce poetry of a similar quality to humans.

To select appropriate human poetry that could be meaningfully compared with the machine-generated poetry, we performed a comprehension test on all poems used in the evaluation, using the Dale-Chall readability formula (Dale and Chall, 1948). This formula represents readability as a function of the complexity of the input words. We selected nine machine-generated poems with a high readability score. The generated poems produced an average score of 7.11, indicating that readers over 15 years of age should easily be able to comprehend them.

For our human poems, we focused explicitly on poetry where greater consideration is placed on prosodic elements like rhythm and rhyme than semantic content (known as "nonsense verse"). We randomly selected 30 poems belonging to that category from the website *poetrysoup.com*, of which

---

[6][URL-ANONYMIZED]

| Poet | Title | Human | Readability | Emotion | Form |
|------|-------|-------|-------------|---------|------|
| **Generated** | **Best** | **0.66** | 0.60 | -0.77 | 0.90 |
| G. M. Hopkins | Carrion Comfort | 0.62 | -1.09 | 1.39 | -1.55 |
| J. Thornton | Delivery of Death | 0.60 | 0.26 | -1.38 | -0.65 |
| **Generated** | **All** | **0.54** | -0.28 | -0.30 | 0.23 |
| M. Yvonne | Intricate Weave | 0.53 | 2.38 | 0.94 | -1.67 |
| E. Dickinson | I'm Nobody | 0.52 | -0.46 | 0.92 | 0.44 |
| G. M. Hopkins | The Silver Jubilee | 0.52 | 0.71 | -0.33 | 0.65 |
| R. Dryden | Mac Flecknoe | 0.51 | -0.01 | 0.35 | -0.78 |
| A. Tennyson | Beautiful City | 0.48 | -1.05 | 0.97 | -1.26 |
| W. Shakespeare | A Fairy Song | 0.45 | 0.65 | 1.30 | 1.18 |

Table 3: Proportion of people classifying each poem as 'human', as well as the relative qualitative scores of each poem as deviations from the mean.

eight were selected for the final comparison based on their comparable readability score. The selected poems were segmented into passages of between four and six lines, to match the length of the generated poetry segments. An example of such a segment is shown in Figure 4d. The human poems had an average score of 7.52, requiring a similar level of English aptitude to the generated texts.

The performance of each human poem, alongside the aggregated scores of the generated poems, is illustrated in Table 3. For the *human* poems, our group of participants guessed correctly that they were human $51.4\%$ of the time. For the *generated* poems, our participants guessed correctly $46.2\%$ of the time that they were machine generated. To determine whether our results were statistically significant, we performed a $Chi^2$ test. This resulted in a $p$-value of $0.718$. This indicates that our participants were unable to tell the difference between human and generated poetry in any significant way. Although our participants generally considered the human poems to be of marginally higher quality than our generated poetry, they were unable to effectively distinguish between them. Interestingly, our results seem to suggest that our participants consider the generated poems to be more 'human-like' than those actually written by humans. Furthermore, the poem with the highest overall quality rating is a machine generated one. This shows that our approach was effective at generating high-quality rhythmic verse.

It should be noted that the poems that were most 'human-like', most aesthetic and most emotive respectively (though not the most readable) were all generated by the neural character model. Generally the set of poetry produced by the neural character model was slightly less readable and emotive than the human poetry, but had above average form. All generated poems included in this evaluation can be found in the supplementary material.

# 6 Conclusions

Our contributions are twofold. First, we developed a neural language model trained on a phonetic transliteration of poetic form and content. Although example output looked promising, this model was limited by its inability to generalise to novel forms of verse. We then proposed a more robust model trained on unformed poetic text, whose output form is constrained at sample time. This approach offers greater control over the style of the generated poetry than the earlier method, and facilitates themes and poetic devices.

An indistinguishability test, where participants were asked to classify a randomly selected set of human "nonsense verse" and machine-generated poetry, showed generated poetry to be indistinguishable from that written by humans. In addition, the poems that were deemed most 'human-like', most aesthetic and most emotive, respectively, were all machine-generated.

In future work, it would be useful to investigate models based on morphemes, rather than characters, which offers potentially superior performance for complex and rare words (Luong et al., 2013), which are common in poetry.

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
