# Peer review of "Automatically Generating Rhythmic Verse with Neural Networks"

_ACL 2017 — decision unknown_

[Official Review · Reviewer 1 · rating 4 · confidence 3]
soundness 3 · originality 4 · clarity 4 · impact 3 · substance 5 · appropriateness 5 · meaningful comparison 5 · presentation format Oral Presentation

The paper presents two approaches for generating English poetry. The first
approach combine a neural phonetic encoder predicting the next phoneme with a
phonetic-orthographic HMM decoder computing the most likely word corresponding
to a sequence of phonemes. The second approach combines a character language
model with a weigthed FST to impose rythm constraints on the output of the
language model. For the second approach, the authors also present a heuristic
approach which permit constraining the generated poem according to theme (e.g;,
love) or poetic devices (e.g., alliteration). The generated poems are evaluated
both instrinsically by comparing the rythm of the generated lines with a gold
standard and extrinsically by asking 70 human evaluators to (i) determine
whether the poem was written by a human or a machine and (ii) rate poems wrt to
readability, form and evocation.  The results indicate that the second model
performs best and that human evaluators find it difficult to distinguish
between human written and machine generated poems.

This is an interesting, clearly written article with novel ideas (two different
models for poetry generation, one based on a phonetic language model the other
on a character LM) and convincing results.

 For the evaluation, more precision about the evaluators and the protocol would
be good. Did all evaluators evaluate all poems and if not how many judgments
were collected for each poem for each task ? You mention 9 non English native
speakers. Poems are notoriously hard to read. How fluent were these ? 

In the second model (character based), perhaps I missed it, but do you have a
mechanism to avoid generating non words ? If not, how frequent are non words in
the generated poems ?

In the first model, why use an HMM to transliterate from phonetic to an
orhographic representation rather than a CRF? 

Since overall, you rule out the first model as a good generic model for
generating poetry, it might have been more interesting to spend less space on
that model and more on the evaluation of the second model. In particular, I
would have been interested in a more detailed discussion of the impact of the
heuristic you use to constrain theme or poetic devices. How do these impact
evaluation results ? Could they be combined to jointly constrain theme and
poetic devices ? 

The combination of a neural mode with a WFST is reminiscent of the following
paper which combine character based neural model to generate from dialog acts
with an WFST to avoid generating non words. YOu should relate your work to
theirs and cite them. 

Natural Language Generation through Character-Based RNNs with Finite-State
Prior Knowledge
Goyal, Raghav and Dymetman, Marc and Gaussier, Eric and LIG, Uni
COLING 2016

[Official Review · Reviewer 2 · rating 3 · confidence 4]
soundness 3 · originality 4 · clarity 3 · impact 3 · substance 4 · appropriateness 5 · meaningful comparison 5 · presentation format Poster

The paper describes two methodologies for the automatic generation of rhythmic
poetry. Both rely on neural networks, but the second one allows for better
control of form.

- Strengths:

Good procedure for generating rhythmic poetry.

Proposals for adding control of theme and poetic devices (alliteration,
consonance, asonance).

Strong results in evaluation of rhythm.

- Weaknesses:

Poor coverage of existing literature on poetry generation.

No comparison with existing approaches to poetry generation.

No evaluation of results on theme and poetic devices.

- General Discussion:

The introduction describes the problem of poetry generation as divided into two
subtasks: the problem of content (the poem's semantics) and the problem of form
(the 

aesthetic rules the poem follows). The solutions proposed in the paper address
both of these subtasks in a limited fashion. They rely on neural networks
trained over corpora 

of poetry (represented at the phonetic or character level, depending on the
solution) to encode the linguistic continuity of the outputs. This does indeed
ensure that the 

outputs resemble meaningful text. To say that this is equivalent to having
found a way of providing the poem with appropriate semantics would be an
overstatement. The 

problem of form can be said to be addressed for the case of rhythm, and partial
solutions are proposed for some poetic devices. Aspects of form concerned with
structure at a 

larger scale (stanzas and rhyme schemes) remain beyond the proposed solutions.
Nevertheless, the paper constitutes a valuable effort in the advancement of
poetry generation.

The review of related work provided in section 2 is very poor. It does not even
cover the set of previous efforts that the authors themselves consider worth
mentioning in their paper (the work of Manurung et al 2000 and Misztal and
Indurkhya 2014 is cited later in the paper - page 4 - but it is not placed in
section 2 with respect to the other authors mentioned there).

A related research effort of particular relevance that the authors should
consider is:

- Gabriele Barbieri, François Pachet, Pierre Roy, and Mirko Degli Esposti.
2012. Markov constraints for generating lyrics with style. In Proceedings of
the 20th European Conference on Artificial Intelligence (ECAI'12), Luc De
Raedt, Christian Bessiere, Didier Dubois, Patrick Doherty, and Paolo Frasconi
(Eds.). IOS Press, Amsterdam, The Netherlands, The Netherlands, 115-120. DOI:
https://doi.org/10.3233/978-1-61499-098-7-115

This work addresses very similar problems to those discussed in the present
paper (n-gram based generation and the problem of driving generation process
with additional constraints). The authors should include a review of this work
and discuss the similarities and differences with their own.

Another research effort that is related to what the authors are attempting (and
has bearing on their evaluation process) is:

- Stephen McGregor, Matthew Purver and Geraint Wiggins, Process Based
Evaluation of Computer Generated Poetry,  in: Proceedings of the INLG 2016
Workshop on Computational Creativity and Natural Language Generation, pages
51–60,Edinburgh, September 2016.c2016 Association for Computational
Linguistics

This work is also similar to the current effort in that it models language
initially at a phonological level, but considers a word n-gram level
superimposed on that, and also features a layer representint sentiment. Some of
the considerations McGregor et al make on evaluation of computer generated
poetry are also relevant for the extrinsic evaluation described in the present
paper.

Another work that I believe should be considered is:

- "Generating Topical Poetry" (M. Ghazvininejad, X. Shi, Y. Choi, and K.
Knight), Proc. EMNLP, 2016.

This work generates iambic pentameter by combining finite-state machinery with
deep learning. It would be interesting to see how the proposal in the current
paper constrasts with this particular approach.

Although less relevant to the present paper, the authors should consider
extending their classification of poetry generation systems (they mention
rule-based expert systems and statistical approaches) to include evolutionary
solutions. They already mention in their paper the work of Manurung, which is
evolutionary in nature, operating over TAG grammars.

In any case, the paper as it stands holds little to no effort of comparison to
prior approaches to poetry generation. The authors should make an effort to
contextualise their work with respect to previous efforts, specially in the
case were similar problems are being addressed (Barbieri et al, 2012) or
similar methods are being applied (Ghazvininejad,  et al, 2016).